

# Spatial habitat suitability prediction of essential oil wild plants on Indonesia's degraded lands

Elga Renjana[1], Elok Rifqi Firdiana[1], Melisnawati H. Angio[1],
Linda Wige Ningrum[2], Intani Quarta Lailaty[1], Apriyono Rahadiantoro[1],
Irfan Martiansyah[1], Rizmoon Zulkarnaen[1,3], Ayyu Rahayu[1],
Puguh Dwi Raharjo[4], Ilham Kurnia Abywijaya[2], Didi Usmadi[2],
Rosniati Apriani Risna[1,5], Wendell P. Cropper, Jr[6] and Angga Yudaputra[2]

[1] Research Center for Applied Botany, National Research and Innovation Agency, Republic of Indonesia, Bogor, West Java, Indonesia
[2] Research Center for Ecology and Ethnobiology, National Research and Innovation Agency, Republic of Indonesia, Bogor, West Java, Indonesia
[3] Faculty of Science, Universiti Brunei Darussalam, Tungku Link, Gadong, Brunei Darussalam
[4] Research Center for Geological Resources, National Research and Innovation Agency, Republic of Indonesia, Bandung, West Java, Indonesia
[5] Natural Resources and Environmental Management Sciences, Bogor Institute of Agriculture, Bogor, West Java, Indonesia
[6] School of Forest, Fisheries and Geomatics Sciences, University of Florida, Gainesville, FL, United States of America

Corresponding author
Elga Renjana, elgarenjana@gmail.com

## ABSTRACT

**Background**. Essential oils are natural products of aromatic plants with numerous uses. Essential oils have been traded worldwide and utilized in various industries. Indonesia is the sixth largest essential oil producing country, but land degradation is a risk to the continuing extraction and utilization of natural products. Production of essential oil plants on degraded lands is a potential strategy to mitigate this risk. This study aimed to identify degraded lands in Indonesia that could be suitable habitats for five wild native essential oil producing plants, namely *Acronychia pedunculata* (L.) Miq., *Baeckea frutescens* L., *Cynometra cauliflora* L., *Magnolia montana* (Blume) Figlar, and *Magnolia sumatrana* var. *glauca* (Blume) Figlar & Noot using various species distribution models.
**Methods**. The habitat suitability of these species was predicted by comparing ten species distribution models, including Bioclim, classification and regression trees (CART), flexible discriminant analysis (FDA), Maxlike, boosted regression trees (BRT), multivariate adaptive regression splines (MARS), generalized linear models (GLM), Ranger, support vector machine (SVM), and Random Forests (RF). Bioclimatic, topographic and soil variables were used as the predictors of the model habitat suitability. The models were evaluated according to their AUC and TSS metrics. Model selection was based on ranking performance. The total suitable area for five native essential oil producing plants in Indonesia's degraded lands was derived by overlaying the models with degraded land locations.
**Results**. The habitat suitability model for these species was well predicted with an AUC value >0.8 and a TSS value >0.7. The most important predictor variables affecting the habitat suitability of these species are mean temperature of wettest quarter, precipitation seasonality, precipitation of warmest quarter, precipitation of coldest quarter, cation
exchange capacity, nitrogen, sand, and soil organic carbon. *C. cauliflora* has the largest predicted suitable area, followed by *M. montana*, *B. frutescens*, *M. sumatrana* var. *glauca*, and *A. pedunculata*. The overlapping area between predictive habitat suitability and degraded lands indicates that the majority of degraded lands in Indonesia's forest areas are suitable for those species.

**Conclusion**. The degraded lands predicted as suitable habitats for five native essential oil producing plants were widely spread throughout Indonesia, mostly in its main islands. These findings can be used by the Indonesian Government for evaluating policies for degraded land utilization and restorations that can enhance the lands' productivity.

# INTRODUCTION

Essential oils are volatile compounds biosynthesized in secretory structures of aromatic plants including cavities, ducts, glands, and hairs located in several parts of aromatic plants (leaves, barks, flowers, fruits, seeds, roots, *etc.*) (*Goodger et al., 2010*; *Goodger et al., 2016*). They are precious liquids, traded commercially worldwide and utilized in various products, including beverages, cosmetics, food items, perfumes, and pharmaceuticals (*Panikar et al., 2021*; *Barbieri & Borsotto, 2018*).

As a global biodiversity hotspot, Indonesia is a natural habitat of many aromatic plant species. Approximately 40 types of essential oils have been produced in Indonesia and some of them have been commercially developed on an industrial scale, *i.e.,* cajuput, ylang-ylang flower, patchouli, *etc.* In addition, Indonesia supplies approximately 90% of the world market demand of patchouli oil and its high quality is widely recognized (*TRECYDA, 2011*). Indonesia has been one of the top six exporters of essential oils, with 7.54 thousand tons valued at 215.19 million USD exported in 2019 (*ITC, 2022*). Therefore, maintaining the sustainable yield of essential oil producing plants in Indonesia has significant implications for economic and community development. A potential major constraint to meeting market demand will be the availability of land fit to produce essential oil raw materials.

Land degradation in Indonesia is a risk factor for sustainable natural product production. In 2018, approximately 14.6 million hectares of degraded lands were reported scattered across Indonesia and attributed to several factors (illegal logging, mining activities, *etc.*). Degradation may be due to high erosion rates and poor soil fertility (*MoFE, 2021*). Since 2010, the Indonesian Government has implemented a policy to mitigate the extent of land degradation through utilizing degraded lands for biomass production (*Gingold et al., 2012*). This activity provides a promising solution for the energy and food security issues, carbon emissions, land restoration, and also to reduce the rate of biodiversity loss (*Borchard et al., 2017*; *Jaung et al., 2018*; *Mujiyo, Widijanto & Herawati, 2022*; *Rahman et al., 2019*).

Consistent with the intent of this Government policy, here we provide a preliminary study regarding degraded land utilization for biomass production, particularly for essential

oil producing plants. We selected five Indonesia native plants with a reasonable chance to survive in harsh conditions. These plants produce essential oils, but they are not currently targeted for harvest. These species should be investigated for potential medicinal and other applications of their essential oil. Those plants include *Acronychia pedunculata* (L.) Miq. *Baeckea frutescens* L., *Cynometra cauliflora* L., *Magnolia montana* (Blume) Figlar, and *Magnolia sumatrana* var. *glauca* (Blume) Fligar & Noot. Leaves and the aerial parts of *A. pedunculata* contain a high concentration of $\alpha$-pinene, caryophyllene, $\beta$-ocimene, globulol (*Lesueur et al., 2008*; *Van et al., 2021*) which have anti-microbial activity against *Salmonella enterica* and *Staphylococcus epidermidis* (*Lesueur et al., 2008*). *B. frutescens*' leaves contain $\alpha$-pinene, $\alpha$-thujene, $\beta$-caryophyllene, linalool, eucalyptol, *etc.* (*An et al., 2020*; *Wang et al., 2019*). One of the compounds, $\beta$-caryophyllene, shows anti-inflammatory action by inhibiting the main mediators of the inflammation process. Furthermore, it may also have applications for treatment of various inflammatory pathologies, such as nervous system diseases, atherosclerosis, and diverse types of cancer (*Francomano et al., 2019*). Essential oils including $\alpha$-terpineol, (Z)-$\beta$-ocimene, and longipinanol can be found in high concentrations in leaves, twigs, and fruits of *C. cauliflora* and show anti-microbial, anti-oxidant, and anti-proliferative activities (*Samling et al., 2021*). Asaricin (sarisan) and safrole are two dominant essential oils in *M. montana*. The former compound has been demonstrated to act as an anti-depressant in tested animals while the latter has anti-bacterial activity against *S. typhimurium* and *Pseudomonas aeruginosa* (*Van Genderen et al., 1999*). The essential oils of *M. sumatrana* var. *glauca* extracted from its roots, branches, leaves and bark are chemically characterized as trans-cinnamaldehyde and caryophyllene (*Wu et al., 2023*). Both compounds have therapeutic effects on cancer and induce apoptosis (*Fidyt et al., 2016*; *Zhang et al., 2015*).

Recently, several studies used spatial approaches to characterize degraded lands in Indonesia that potentially could become more productive lands (*Artati et al., 2019*; *Gingold et al., 2012*; *Jaung et al., 2018*). The suitability of degraded lands for oil palm cultivation in Central and West Kalimantan was mapped using several indicators (land cover, erosion risk, elevation, slope, *etc.*) (*Gingold et al., 2012*). In addition, approximately 3.5 million hectares of degraded lands in Indonesia were estimated to be suitable for biodiesel production species (*Calophyllum inophyllum*, *Pongamia pinnata* and *Reutealis trisperma*) and biomass species (*Calliandra calothyrsus* and *Gliricidia sepium*) (*Jaung et al., 2018*). However, there has been no research concerning the suitability potential of degraded lands in Indonesia for essential oil producing plants. Therefore, in this study, we compared ten machine learnings including Bioclim, classification and regression trees (CART), flexible discriminant analysis (FDA), Maxlike, Boosted Regression Trees (BRT), multivariate adaptive regression splines (MARS), generalized linear models (GLM), Ranger, support vector machines (SVM), and Random Forests (RF) to identify an accurate and robust model to identify degraded lands suitable for each species and to evaluate conservation implications of employing this strategy.

## MATERIALS & METHODS

### Data preparation

The occurrence records of five native essential oil producing plants were obtained from field surveys, the GBIF database (*GBIF, 2023a*; *GBIF, 2023b*; *GBIF, 2023c*; *GBIF, 2023d*; *GBIF, 2023e*), and the iNaturalist database (*iNaturalist, 2024*). A number of occurrence records that we identified for *A. pedunculata*, *B. frutescens*, *C. cauliflora*, *M. montana*, and *M. sumatrana* var. *glauca* were 35, 44, 28, 42, and 32 respectively (Fig. 1). The map of Indonesia's degraded lands in 2022 with five categories (undegraded, moderately degraded, potentially degraded, degraded, and highly degraded) was derived from the Ministry of Environment and Forestry, the Republic of Indonesia. In this study, we focused on the degraded and highly degraded categories (Fig. 2). In addition, degraded lands classified as agricultural, mining, residential areas, water bodies, swamps, ponds and air/sea ports were excluded by overlaying the Indonesia's degraded land map with the Indonesia's land cover map 2022 from the Ministry of Environment and Forestry.

Fifteen variables consisting of bioclimatic, topographic and soil variables were used as predictors to model habitat suitability. These predictors were selected according to the results of a multicollinearity test using the VIFcor function/analysis with a threshold of 0.7, available in the Uncertainty Analysis for Species Distribution Models (usdm) R package version 2.1-7 (*Dormann et al., 2013*). The bioclimatic data consisted of temperature seasonality, mean temperature of the wettest quarter, precipitation of the wettest month, precipitation seasonality, and precipitation of the warmest quarter, and precipitation of the coldest quarter. Those data were available at 30 arc s ($\sim$1 km$^2$) and extracted from WorldClim version 2.1 (*Fick & Hijmans, 2017*; https://www.worldclim.org/). The topography data consisted of aspect and slope which were estimated from elevation data. The elevation data with a 1 arc second resolution ($\sim$30 m) was derived from the Shuttle Radar Topography Mission (SRTM) Digital Elevation Model Data (*Farr et al., 2007*; *NASA, 2013*). Soil data consisted of chemical characteristics (cation exchange capacity, nitrogen content, soil pH, and soil organic carbon) and physical characteristics (bulk density, clay content, coarse fragments, and sand) in the 30–60 cm depth range. Those data were derived from SoilGrids (a system for global digital soil mapping) with a 250 m resolution (*Hengl et al., 2017*; https://soilgrids.org/). All predictor data were adjusted into similar spatial resolution ($\sim$1 km$^2$).

### Species distribution modeling

The habitat suitability of the five native essential oil producing plants were predicted using 10 algorithms in the Species Distribution Modelling (sdm) R package version 1.1-8: Bioclim (*Booth et al., 2014*), CART (*Loh, 2011*), FDA (*Karami et al., 2021*), Maxlike (*Royle et al., 2012*), BRT (*Elith, Leathwick & Hastie, 2008*), MARS (*Elith & Leathwick, 2007*), GLM (*Hardin & Hilbe, 2007*), Ranger (*Wright & Ziegler, 2017*), SVM (*Noble, 2006*), and RF (*Breiman, 2001*). All algorithms were run simultaneously for each species. The models were built using two different runs of subsampling and bootstrap replications. About 25% of the data were designated as testing data used for analyzing out-of-sample model performance. A previous study revealed that a data split of 75% for model building and 25% for model

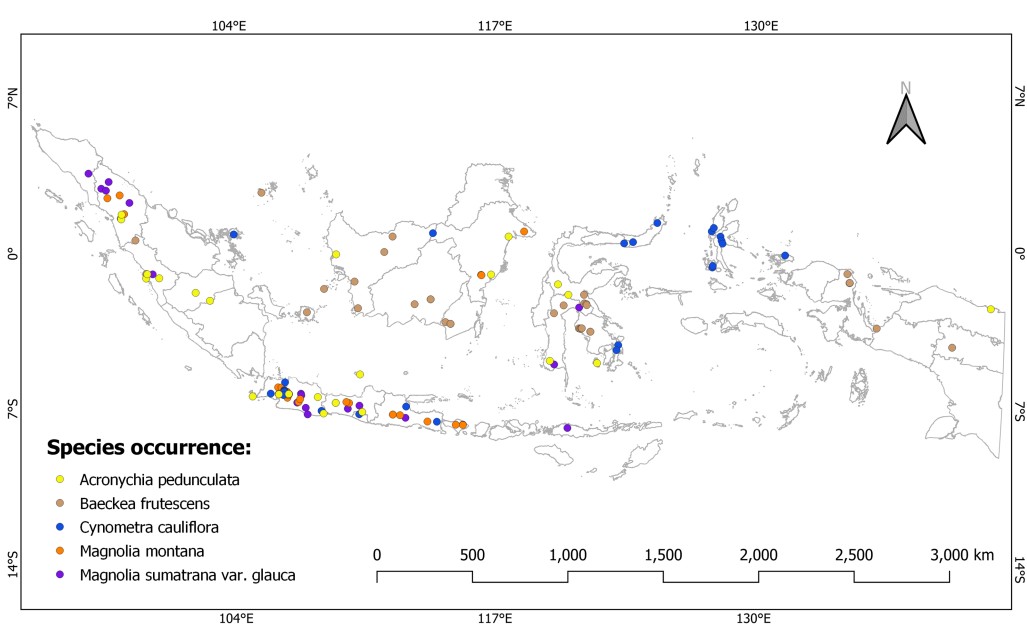

**Figure 1** **Occurrence records of selected wild plants containing essential oils.** Coordinates source: field survey, GBIF.org, and iNaturalist.org; processed further using QGIS 3.18.

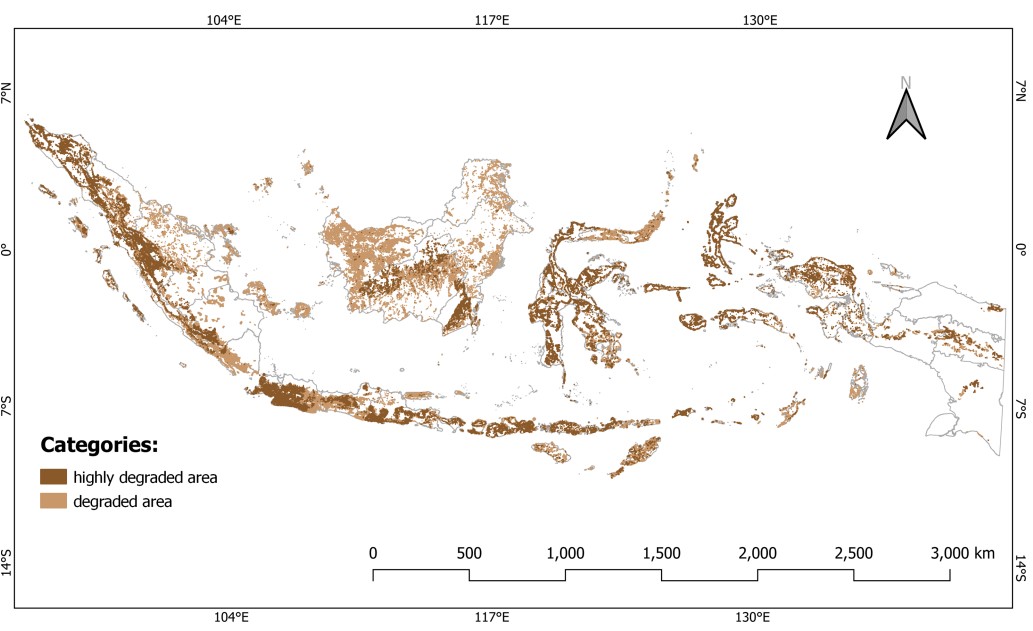

**Figure 2** **Map of Indonesia's degraded lands in 2022.** Source: Ministry of Environment and Forestry, the Republic of Indonesia; processed further using QGIS 3.18.

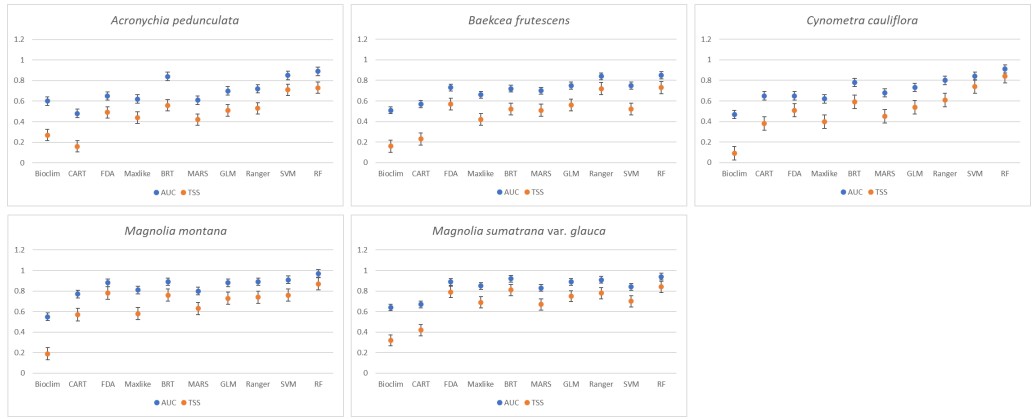

**Figure 3  The performance of habitat suitability models of wild plant species containing essential oils.**

validation, resulted in higher values of AUC compared to other splitting data criteria (*Yudaputra, Astuti & Cropper, 2019*). The model performance was evaluated using the area under the receiver operating characteristic curve (AUC) and the true skill statistics (TSS) metrics.

## Determination of total degraded land area suitable for native essential oil producing plants

The predictive suitability maps were reclassified into four classes of potential habitat as follows: unsuitable ($\leq 0.10$), marginally suitable (0.11–0.30), moderately suitable (0.31–0.70), and highly suitable ($\geq 0.71$) (*Choudhury et al., 2016*; *Qin et al., 2017*; *Yang et al., 2013*). These maps were spatially intersected with Indonesia's degraded land map 2022 and Indonesia's forest area map 2023 using QGIS version 3.18 (*QGIS.org, 2024*) in order to derive the total suitable area for the five native essential oil producing plants in degraded lands.

## RESULTS

Evaluation performance of the 10 machine learning algorithms identified RF as the best predictor of habitat suitability for the five wild plant species containing essential oils. The AUC values of models of *A. pedunculata, B. frutescens, C. cauliflora, M. montana*, and *M. sumatrana* var. *glauca* are 0.89, 0.85, 0.91, 0.97, and 0.94 respectively. While the TSS values of *A. pedunculata, B. frutescens, C. cauliflora, M. montana*, and *M. sumatrana* var. *glauca* are 0.73, 0.73, 0.84, 0.87, 0.84 respectively (Fig. 3).

The maps of model outputs represent the habitat suitability of those species throughout Indonesia (Fig. 4). Java, Sunda Lesser Is., and southern Sulawesi include regions that are predicted to be suitable for *A. pedunculata*. The highly suitable areas for *B. frutescens* are predicted to be in Sulawesi, Moluccas Is., and Papua. Suitable habitat areas of *C. cauliflora* are predicted in most areas in Java, Kalimantan, Sulawesi, Sumatra, Papua, and several areas in Moluccas Is. and Sunda Lesser Is. Similar to *C. cauliflora*, suitable habitat areas of *M. montana* are predicted in all Indonesian main islands, *i.e.,* most regions in Java, the

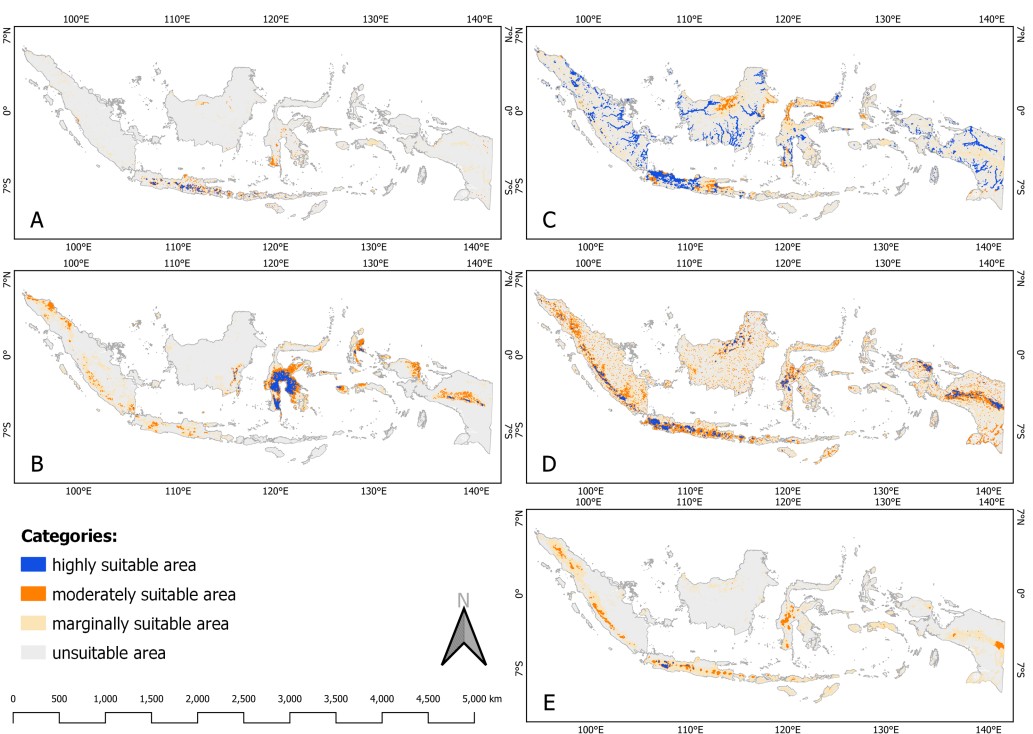

**Figure 4  Predicted suitable habitats of wild plant species containing essential oils provided by Random Forests Models.** (A) *Acronychia pedunculata.* (B) *Baeckea frutescens.* (C) *Cynometra cauliflora.* (D) *Magnolia montana.* (E) *Magnolia sumatrana* var. *glauca.*

southwestern Sumatran highlands, the northern Kalimantan, central Sulawesi, northern and central Papua, several regions of Sunda Lesser Is. and Mollucas Is. Besides that, West and central Java are predicted to be highly suitable areas for *M. sumatrana* var. *glauca.*

*Cynometra cauliflora* has the largest suitable area of 19365.96 km², followed by *M. montana, B. frutescens, M. sumatrana* var. *glauca,* and *A. pedunculata* which cover 5,959.62, 5,900.52, 734.16, and 576.46 km² respectively. Java, Sumatra, Kalimantan, and Papua are predicted to include the largest areas of suitable habitat for *C. cauliflora* with an area of 8317.38, 3372.05, 3038.83, and 2749.01 km². Sulawesi has more suitable habitat for *B. frutescens* with an area of 5,554.19 km², while Java and Papua have the largest suitable area for *M. montana* 3,534.06 and 1,043.95 km² (Table 1).

The three most important variables affecting the habitat suitability of *A. pedunculata* and *M. sumatrana* var. *glauca* are mean temperature of the wettest quarter, soil cation exchange capacity, and precipitation seasonality. Precipitation of warmest quarter, nitrogen content, and soil organic carbon are three important variables for *B. frutescens.* Precipitation of the coldest quarter, nitrogen content, and sand content are three important variables for *C. cauliflora.* The mean temperature of the warmest quarter, sand content, and precipitation of the coldest quarter are the strongest explanatory variables for *M. montana* (Fig. 5).

The overlapping area between predictive habitat suitability and degraded lands (degraded and highly degraded areas) indicates Indonesia's degraded lands that are potentially suitable

**Table 1  Highly suitable habitat of selected wild plant species containing essential oils.**

| Island (s) | Highly suitable habitat (km$^2$) | | | | |
|---|---|---|---|---|---|
| | *A. pedunculata* | *B. frutescens* | *C. cauliflora* | *M. montana* | *M. sumatrana* var. *glauca* |
| Java | 550.83 | 2.88 | 8317.38 | 3534.06 | 734.16 |
| Kalimantan | 0.00 | 23.38 | 3038.83 | 312.81 | 0.00 |
| Lesser Sunda Islands | 22.47 | 0.00 | 110.34 | 65.10 | 0.00 |
| Moluccas Islands | 0.00 | 112.38 | 214.59 | 43.52 | 0.00 |
| Sulawesi | 3.16 | 5554.19 | 797.76 | 285.84 | 0.00 |
| Sumatra | 0.00 | 0.96 | 3372.05 | 654.06 | 0.00 |
| Papua | 0.00 | 206.74 | 2749.01 | 1043.95 | 0.00 |
| Total Area | 576.46 | 5,900.52 | 19,365.96 | 5,959.62 | 734.16 |

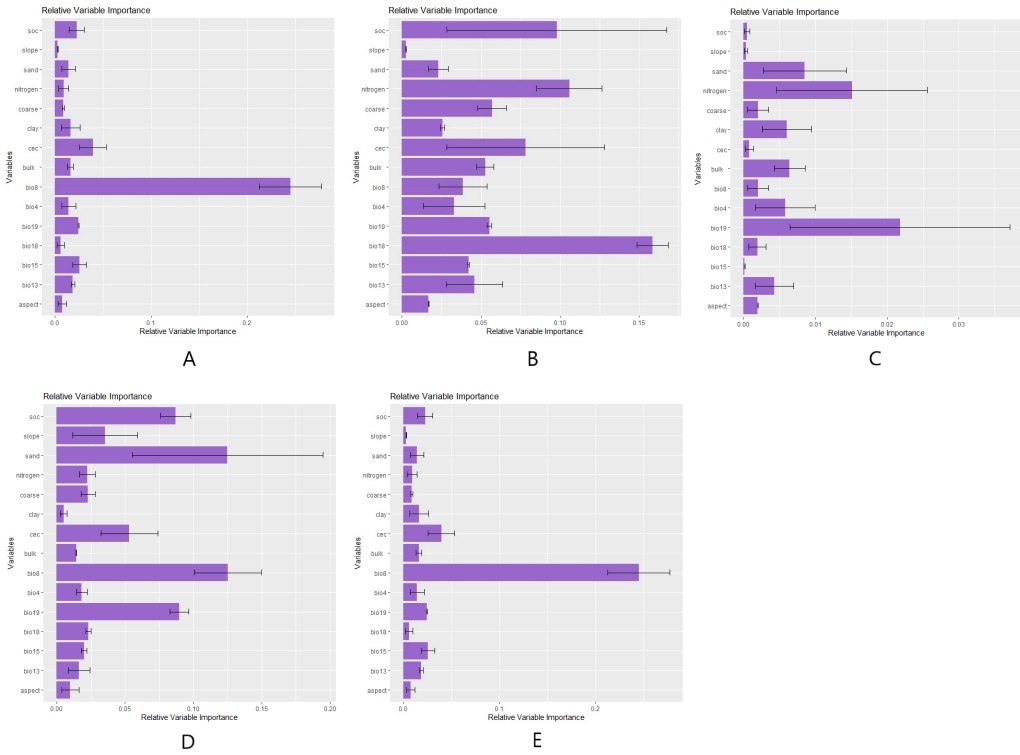

**Figure 5  Importance variables affecting habitat suitability of selected wild plant species containing essential oils.** (A) *Acronychia pedunculata*. (B) *Baeckea frutescens*. (C) *Cynometra cauliflora*. (D) *Magnolia montana*. (E) *Magnolia sumatrana* var. *glauca*. Remarks: bio4 = temperature seasonality, bio8 = mean temperature of wettest quarter, bio13 = precipitation of wettest month, bio15 = precipitation seasonality, bio18 = precipitation of warmest quarter, bio19 = precipitation of coldest quarter, bulk = bulk density, cec = cation exchange capacity, coarse = coarse fragments, soc = soil organic carbon.

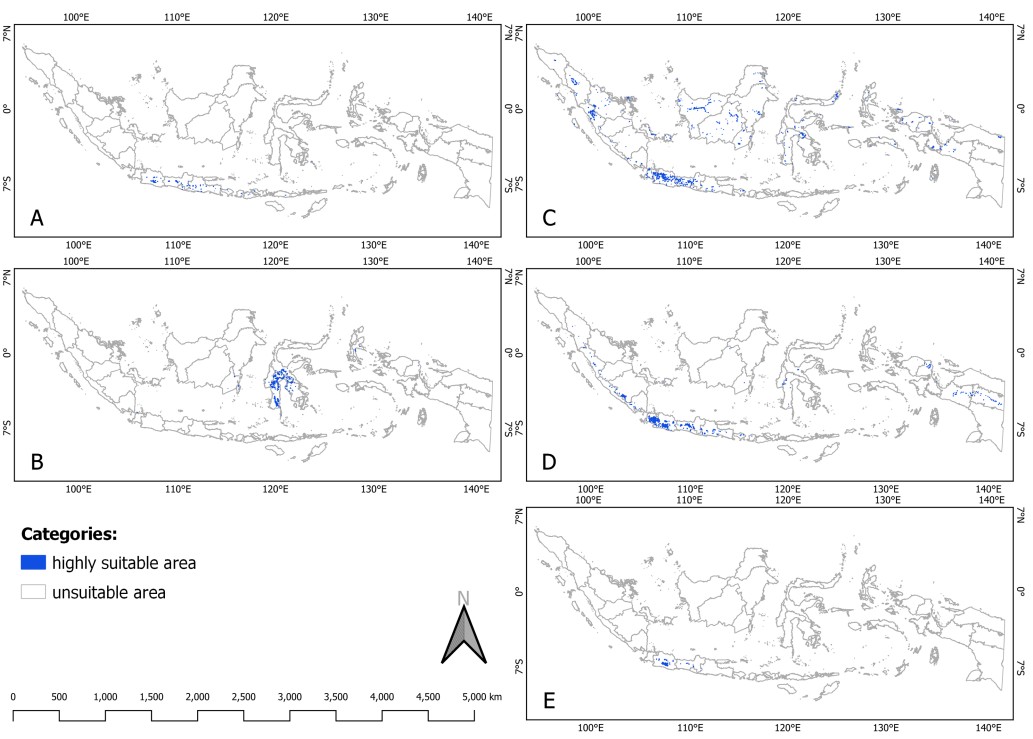

**Figure 6** **The overlapping areas between Indonesia's degraded lands and suitable habitat of five native essential oils producing plants in forest areas.** (A) *Acronychia pedunculata*. (B) *Baeckea frutescens*. (C) *Cynometra cauliflora*. (D) *Magnolia montana*. (E) *Magnolia sumatrana* var. *glauca*.

for the five native essential oil producing plants. These species are estimated to be suitable for the majority of degraded land in forest areas, while suitable habitat in non-forest areas are very small (<0.01 km$^2$), hence considered insignificant (Fig. 6 and Table 2). Numerous scattered areas of degraded lands in Java are highly suitable for all of the studied species (1,392.17 km$^2$), with additional areas of Sulawesi for *B. frutescens* (555.12 km$^2$) and Sumatra for *C. cauliflora* (168.79 km$^2$). The total degraded areas suitable for *A. pedunculata, B. frutescens, C. cauliflora, M. montana* and *M. sumatrana* var. *glauca* are 109.71 km$^2$, 563.08 km$^2$, 554.65 km$^2$, 1,049.92 km$^2$, and 187.88 km$^2$ respectively. In summary, the degraded lands potentially suitable for those species are about 2,465.24 km$^2$ (Table 2).

# DISCUSSION

In this study, the RF models for each of the species studied have AUC values >0.8 and TSS values >0.7 (Fig. 3). The AUC value has a range between 0 and 1, while 0.5 was set as threshold value indicating the model performance is not better than random in distinguishing presence and absence data (*Fan, Upadhye & Worster, 2006*). Models performance can be summarized based on AUC values. AUC values in the range 0.9–1 are considered to be excellent, those with AUC values from 0.8−0.9 are good, those from 0.7−0.8 are fair, those from 0.6−0.7 are poor, and AUC values in the range of 0.5−0.6

**Table 2  Overlapping areas between suitable habitat of five native essential oil producing plants and Indonesia's degraded lands in each province.**

| Provinces in each island | Forest area (km$^2$) | | | | |
|---|---|---|---|---|---|
| | *Ap* | *Bf* | *Cc* | *Mm* | *Ms* |
| **Java** | | | | | |
| Banten | – | 1.86 | 11.63 | 124.99 | – |
| Central Java | 50.54 | – | 58.23 | 184.39 | 4.24 |
| East Java | 10.06 | – | 2.74 | 51.98 | 0.07 |
| West Java | 43.25 | – | 115.42 | 547.86 | 183.57 |
| Yogyakarta Special Region | 0.40 | – | 0.19 | 0.75 | – |
| **Kalimantan** | | | | | |
| Central Kalimantan | – | – | 27.82 | 0.99 | – |
| East Kalimantan | – | – | 11.64 | – | – |
| North Kalimantan | – | – | 0.98 | – | – |
| South Kalimantan | – | 3.58 | 30.15 | 0.43 | – |
| West Kalimantan | – | – | 23.13 | – | – |
| **Sunda Lesser Islands** | | | | | |
| Bali | 1.06 | – | 3.73 | 3.04 | – |
| East Nusa Tenggara | 0.10 | – | 0.07 | – | – |
| West Nusa Tenggara | 3.72 | – | – | 0.37 | – |
| **Moluccas Islands** | | | | | |
| Moluccas | – | – | 2.89 | – | – |
| North Moluccas | – | 2.49 | 4.79 | – | – |
| **Papua** | | | | | |
| Central Papua | – | – | 3.97 | 21.66 | – |
| Highland Papua | – | – | – | 29.75 | – |
| Northwest Papua | – | – | 7.27 | 0.03 | – |
| Papua | – | – | 14.06 | – | – |
| South Papua | – | – | – | – | – |
| West Papua | – | 0.04 | 13.41 | 8.48 | – |
| **Sulawesi** | | | | | |
| Central Sulawesi | – | 25.31 | 8.81 | 4.04 | – |
| Gorontalo | – | – | 0.36 | – | – |
| North Sulawesi | – | – | 15.08 | – | – |
| South Sulawesi | – | 506.73 | 27.75 | 4.54 | – |
| Southeast Sulawesi | 0.64 | 21.14 | 0.57 | – | – |
| West Sulawesi | – | 1.94 | 1.17 | 3.61 | – |
| **Sumatra** | | | | | |
| Bangka Belitung | – | – | 13.63 | 0.21 | – |
| Bengkulu | – | – | 1.58 | 2.03 | – |
| Jambi | – | – | 4.95 | 1.46 | – |
| Lampung | – | – | 14.70 | 10.05 | – |
| Nanggroe Aceh Darussalam | – | – | 6.60 | – | – |
| North Sumatra | – | – | 33.45 | 2.62 | – |

**Table 2** (*continued*)

| Provinces in each island | Forest area (km²) | | | | |
|---|---|---|---|---|---|
| | *Ap* | *Bf* | *Cc* | *Mm* | *Ms* |
| Riau | – | – | 29.65 | – | – |
| Riau Islands | – | – | 8.92 | – | – |
| South Sumatra | – | – | 3.51 | 36.79 | – |
| West Sumatra | – | – | 51.80 | 9.86 | – |
| Total Area | 109.71 | 563.08 | 554.65 | 1,049.92 | 187.88 |

**Notes.**

Ap, A. pedunculata; Bf, B. frutescens; Cc, C. cauliflora; Mm, M. montana; Ms, M. sumatrana var. glauca.

are failed models (*Krzanowski & Hand, 2009*). Meanwhile, the TSS value is based on the components of the standard confusion matrix, which represents matches and mismatches between observations and predictions (*Fielding & Bell, 1997*). It ranges from −1 to 1, while models with a TSS >0.5 are generally acceptable (*Allouche, Tsoar & Kadmon, 2006*). The AUC and TSS values for the essential oils producing plants represent the acceptable categories of predictive model performance. Therefore, this study supports the conclusion that the models are useful predictors of habitat suitability for the plant species.

These predictions are primarily influenced by a small number of bioclimatic and soil parameters (Fig. 5). Temperature is the most important variable in three of the species' distribution models (*A. pedunculata, M. montana,* and *M. sumatrana* var. *glauca*). Every plant species has a range of minimum and maximum temperature for its vegetative and reproductive development. The best plant development usually occurs around the optimum temperature (*Hatfield & Prueger, 2015*). Precipitation is the most importance predictor for *B. frutescencs* and *C. cauliflora*. It influences water availability and the soil moisture status in the ecosystem (*Sun et al., 2021*; *Wu et al., 2011*), and also enhances weathering in the soil that promotes the production of secondary clay minerals (*Ahmad et al., 2018*). Precipitation may also regulates plant gas exchange by influencing plant photosynthesis and transpiration rates (*Song, Niu & Wan, 2016*). These climate variables influence carbon input into the soil, and therefore soil quality, as a result of influencing plant growth activities (*Zhang & Xi, 2021*).

Several soil parameters such as cation exchange capacity, nitrogen content, sand content, and soil organic carbon also are important predictors for the five native essential oil producing plants. Cation exchange capacity is associated with the soil's ability to store and exchange cations, greatly influencing plant nutrition. Cation exchange capacity is affected by a number of soil properties such as soil texture, pH, *etc.* (*Khaledian et al., 2017*). Nitrogen is the most essential element for plant growth and development of all the nutrient compounds required. It plays a key role in most plant metabolic processes (*Fathi, 2022*). The sand content of soils influences the structural stability in the infield soil. The presence of sand in the right quantity and size will create pore spaces and leave room for smaller particles such as silt and clay (*Kollaros & Athanasopoulou, 2017*). Soil organic carbon is the main component of soil organic matter and becomes a crucial factor for ecosystem functions affecting soil fertility to support and sustain plant growth. Its level depends on the equilibrium of carbon inputs and losses in the soil (*FAO, 2017*).

Previous studies have reported that those five species are adapted to grow and develop in harsh environments. In Hong Kong, *A. pedunculata* is widely distributed in degraded hillsides (*Hau & Corlett, 2002*). Additionally, *A. pedunculata* along with *Maesa indica* and *Macaranga* spp. were known as pioneer tree species in Sri Lanka forests. They have resilient seeds and play an important role in facilitating the regeneration process after disturbance (*Millet & Truong, 2011*; *Piyasinghe, Gunatilake & Madawala, 2017*). *Baeckea frutescens*, a wind-dispersed and fire-tolerant species, also grows well on degraded hillsides in Hong Kong (*Hau & Corlett, 2002*). Moreover, it is also able to grow in former quartz sand mining land after five and fifteen years of operations with the important value index (IVI) being 38.01% and 26.67% respectively (*Hilwan, 2015*).

Seedlings of *C. cauliflora* have been experimentally planted on a severely degraded landfill in a Forest Reserve in Selangor, Malaysia. Despite the low macronutrients and organic contents, they showed a relatively high survival rate of 42.4% (*Azani et al., 2005*). Little is currently known about the performance of *M. montana* in degraded areas, but *M. montana* is believed to be beneficial for improving soil and water conservation. *M. montana* is also able to reduce surface runoff, hence preventing soil erosion (*Faye, Setiadi & Qayim, 2011*). *M. sumatrana* var. *glauca* is one of many recommended plants for land restoration due to its leaf photosynthetic related traits, including total chlorophyll, leaf moisture, leaf weight, and leaf area (*Ahmad, Setiadi & Widyatmoko, 2013*). This species can reach maturity in under 10 years, and thus belongs to the fast-growing tree species category (*Sudomo, Rachman & Mindawati, 2010*). In 2009, *M. sumatrana* var. *glauca,* which was planted in degraded lands of Gede Pangrango National Park, Indonesia, had a survival rate of about 90% (*Rahman et al., 2011*).

Land degradation refers to a clear performance reduction in the soil and biota, including various productivity, health, strength, and environmental functions (*Mahala, 2020*). Degradation occurs due to a mismatch between land capacity and land use, resulting in physical, chemical and biological damage to the land (*Arsyad, 2010*). Based on our findings, *A. pedunculata, B. frutescens, C. cauliflora, M. montana* and *M. sumatrana* var. *glauca* are highly suitable for cultivation in Indonesia's degraded lands. Growing essential oil producing plants on degraded land can open up new avenues for sustainability. It is not only increases net productivity, but also restores the soil without chemical intervention (*Yadav et al., 2023*).

Most of Indonesia's degraded lands suitable for those species were located in forest areas. It is important to note that their origin areas must be taken into account before using them in forest conservation efforts. This could imply two advantages at once; the land would be restored and the plants would be conserved *in situ* as well. Fortunately, the predicted suitable habitats on Indonesia's degraded lands for all the species studied are mostly in their native areas. Thus, we recommend *A. pedunculata* and *M. sumatrana* var. *glauca* to restore degraded lands in Java, *B. frutescens* in Sulawesi, *C. cauliflora* in Java, Kalimantan, Sulawesi, and Sumatra, and *M. montana* in Java, the Lesser Sunda Is., and Sumatra.

This study demonstrates the application of a model that identifies degraded lands as suitable habitat for five native essential oil producing plants and characterizes the important environmental variables affecting those species. However, careful consideration

should be applied in making use of the model predictions. Additional information, such as the detailed characteristics of the degraded lands and the plants' various horticultural aspects, should be considered while utilizing degraded lands with these five studied species. Additionally, a small preliminary planting experiment should also be carried out using the essential oil producing plants prior to the large-scale planting in the degraded lands. In the future, these findings could be used by the Indonesian Government in determining policies for degraded land utilization and restoration enhance the lands' productivity.

## CONCLUSIONS

The suitable habitats of five wild native essential oil producing plants are predicted to include scattered degraded lands in forest areas across Indonesia, dominantly in its main islands. Highly suitable habitats for *A. pedunculata, C. cauliflora, M. montana,* and *M. sumatrana* var. *glauca* are mostly predicted in degraded lands in Java. Degraded lands in Sumatra are highly suitable for *C. cauliflora* and degraded lands in Sulawesi for *B. frutescens*. The mean temperature of wettest quarter, precipitation seasonality, precipitation of warmest quarter, precipitation of coldest quarter, cation exchange capacity, soil nitrogen content, sand content, and soil organic carbon are the important variables in predicting these species' suitable habitat. To ensure conservation efforts are effective, it is important to consider the original distribution area of each species when determining the location of degraded land with high habitat suitability.

## ACKNOWLEDGEMENTS

We deliver our great appreciation to the Ministry of Environment and Forestry, the Republic of Indonesia for providing the degraded land, forest area, and land cover maps.

### Funding
The authors received no funding for this work.

### Competing Interests
The authors declare there are no competing interests.

### Author Contributions
- Elga Renjana conceived and designed the experiments, performed the experiments, analyzed the data, prepared figures and/or tables, authored or reviewed drafts of the article, and approved the final draft.
- Elok Rifqi Firdiana performed the experiments, authored or reviewed drafts of the article, and approved the final draft.
- Melisnawati H. Angio performed the experiments, authored or reviewed drafts of the article, and approved the final draft.
- Linda Wige Ningrum performed the experiments, authored or reviewed drafts of the article, and approved the final draft.

- Intani Quarta Lailaty performed the experiments, authored or reviewed drafts of the article, and approved the final draft.
- Apriyono Rahadiantoro performed the experiments, authored or reviewed drafts of the article, and approved the final draft.
- Irfan Martiansyah performed the experiments, authored or reviewed drafts of the article, and approved the final draft.
- Rizmoon Zulkarnaen performed the experiments, authored or reviewed drafts of the article, and approved the final draft.
- Ayyu Rahayu performed the experiments, authored or reviewed drafts of the article, and approved the final draft.
- Puguh Dwi Raharjo performed the experiments, prepared figures and/or tables, authored or reviewed drafts of the article, and approved the final draft.
- Ilham Kurnia Abywijaya performed the experiments, authored or reviewed drafts of the article, and approved the final draft.
- Didi Usmadi analyzed the data, prepared figures and/or tables, authored or reviewed drafts of the article, and approved the final draft.
- Rosniati Apriani Risna performed the experiments, authored or reviewed drafts of the article, and approved the final draft.
- Wendell P. Cropper, Jr analyzed the data, authored or reviewed drafts of the article, and approved the final draft.
- Angga Yudaputra conceived and designed the experiments, performed the experiments, analyzed the data, prepared figures and/or tables, authored or reviewed drafts of the article, and approved the final draft.

## Data Availability

The data is available at FigShare: Renjana, Elga (2024). Essential Oils Data. figshare. Dataset. https://doi.org/10.6084/m9.figshare.25211531.v1.

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
