# Peer review of "Spatial habitat suitability prediction of essential oil wild plants on Indonesia's degraded lands"

_PeerJ, doi:10.7717/peerj.17210_

## Round 0.1 · original submission · Major Revisions

Despite the potentialities of this manuscript, both reviewers agreed to identify major issues and shortcomings in your paper. In particular, they suggested to improve the introduction, results and discussion sections. Furthermore, since the recent application of essential oils in the cosmetic, pharmaceutical industry, and agriculture field, in addition to the reviewer comments, I would like to urge you to develop the introduction section underlining the application potentialities of the five essential oil wild plants in these fields. When revising your manuscript, please consider all issues mentioned in the reviewers' comments carefully: please outline every change made in response to their comments and provide suitable rebuttals for any comments not addressed.

·

Basic reporting

I have read with great interest the manuscript of Renjana et al. with the title “Predicting habitat suitability of essential oil wild plants on Indonesia’s degraded lands using maximum entropy”, and I found it to be of considerable validity from the methodological point of view, but it is not clear in terms of its general purpose, whether it emphasizes the potential aspect of using degraded land for the cultivation of the five oil wild plants species or the implications important in conserving five oil wild plant species in the future. In particular, the manuscript explores the effect of various factors on the distribution of five oil wild plants on Indonesia’s degraded lands using the Maximum Entropy (Maxent) model.

Besides, I feel that the emphasis should be placed on the results with ecological and conservation implications, not on the technical aspects concerning the performance of the Maxent model. I think this should be discussed in more detail in the text. Therefore, I would suggest restructuring the results and discussion parts, bringing forward the potential distribution of the five oil wild plants and the effects of the considered drivers revealed by the best models, and addressing the technical issues afterward and the justification for choosing the Maxent model.

Overall, the manuscript is easy to follow and concise, but there are a few grammar issues (some identified below), so I recommend revising these issues. For example, in the section of Abstract: Line 46 (change "degrades" to "degraded"), Line 46 (change "as" to "to be"), Line 48 (change “to be great aid” to (to greatly aid), etc.

Experimental design

I have two critical points that the authors should address:
1) The multimodel approach is certainly very useful for identifying the potential presence of species starting from presence-only data. It is important to choose the model with the best performance. Various research showed that the Maxent was not the top-performing model, compared with other models such as Random Forest (RF), Ensembles, etc (e.g., Oppel et al. 2012; Rahman et al. 2022; Gao et al. 2023).

• Oppel S, Meirinho A, Ramírez I, Gardner B, O’Connell A.F, Miller P.I, Louzao M. 2012. Comparison of five modelling techniques to predict the spatial distribution and abundance of seabirds. Biological Conservation 156, 94-104. https://doi.org/10.1016/j.biocon.2011.11.013
• Rahman D.A, Santosa Y, Purnamasari I, Condro A.A. 2022. Drivers of Three Most Charismatic Mammalian Species Distribution across a Multiple-Use Tropical Forest Landscape of Sumatra, Indonesia. Animals 12, 2722. https://doi.org/10.3390/ani12192722
• Gao C, Hong Y, Sun S, Zhang N, Liu X, Wang Z, Zhou S, Zhang M. 2023. An Evaluation of Suitable Habitats for Amur Tigers (Panthera tigris altaica) in Northeastern China Based on the Random Forest Model. Biology 12, 1444. https://doi.org/10.3390/biology12111444
2) It would be necessary to have more details on the statistical analyses and the models used.

Validity of the findings

Reading the manuscript, the authors tend only to evaluate the factors associated with the distribution of the five oil wild plants species using one model. Moreover, the discussion might be too limited. Although interesting results are given, this may be of limited benefit when not linked to the future of the five oil wild plants species management aspects and the utilization of the Maxent model and their results for the five oil wild plants species conservation. Please add at the end of the discussion regarding "Habitat Conservation and Management Recommendations for the five oil wild plants species" in relation to the main research findings.

Additional comments

Lines 32-33: Why did the author choose these five species in their study? The author should explain the choice of this species from the beginning, for example, because of its high utility value or the conservation status. I didn't find any explanation in the Introduction section, and it suddenly appeared in the Methods section.
Lines 39-40: AUC has received criticism from researchers (Lobo et al. 2010, Lobo & Tognelli 2011) because dependence on prevalence (i.e., the proportion of a species' range present) leads to bias in the metric. The author used AUC values to evaluate the model performance, and the results were very high (0.955). There is a lot of literature regarding limitations and various explanations which show that an AUC value that is too high does not necessarily indicate a prediction model with high accuracy. For example, AUC values tend to be higher for species with narrow distributions or little data, so high AUC values do not always describe a good model; this is a value that is built (artifact) from the results of analysis of AUC statistics (Phillips et al. 2006; Phillips & Dudik 2008). I recommend that you use discriminant matrices such as Kappa, true skill statistics (TSS), Jaccard, and Sørensen (see Allouche et al. 2006; Leroy et al. 2018; Rahman et al. 2022) in Species distribution modeling Section (Line 181-186).
Lines 61-62: Do all traditional herbal medicines fall into the category or product known as jamu (in Indonesia)?
Line 65-68: It is best to provide examples of medical uses of native plants that relate to the use of the five species you have selected in this manuscript.
Lines 70-71: Add reference.
Lines 71-73: Add reference.
Lines 82-83: Why use data 2018 for land degradation in your research? I suggest the author use the latest land degradation data to show current conditions.
Lines 100-101: The author is using a Maxent model! Why use this model when there are still many better algorithm models like Random Forest and Ensembles? This manuscript should have 1 paragraph or two/three sentences discussing the selection of Maxent models. Furthermore, I suggest that the authors use a more reliable SDM model with a better level of precision, such as Random Forest and Ensembles.
Lines 100-101: What is the hypothesis of this research?
Lines 105-106: I guess the occurrence data the author uses comes entirely from metadata from the GBIF database. How with primary data from the field surveys or data from other databases? Considering the very limited number of occurrence data in your research, I suggest the author add other occurrence data from various sources, considering that you are modeling the spatial distribution of the five species over a large area (across Indonesia).
Line 108: See my comments in the Introduction regarding this.
Line 118: What is your basis for selecting the three environmental variables to model the distribution of the five species? The accuracy of selecting environmental variables largely determines the strength of a species spatial model.
Lines 132-167: This description of species should be elaborated on in the Introduction section. See my comments in Lines 32-33.
Lines 174-177: How do you process it? Are all species run simultaneously or one by one?
Lines 181-186: See my comments in Lines 39-40.
Lines 196-197: The authors show that the AUC values for the entire model are very high. See my comments and consent regarding the weakness of the AUC Values in Lines 39-40.
Lines 336-345: The discussion is well structured but largely focused on the variable influence. To better reach the international audience, the authors should discuss and emphasize, based on their findings, what are the contributions of this study on a broader scale, e.g., for other species in other areas. The authors also need to elaborate more on what are the implications of the conservation management of species and not only emphasize the potential of their cultivation on degraded land.
Lines 343-345: How large an area is defined as the suitable habitat for each species? Given that you are addressing the future use and restoration of degraded land for all five species, information on how large an area is defined as a suitable habitat is essential in future planning.

Reviewer 2 ·

Basic reporting

The manuscript is well written, so it is very easy to understand

Experimental design

a. degraded land data from the Ministry of Environment and Forestry 2018 has been updated in 2022, if possible it can be updated!

b. Critical land on the Ministry of Environment and Forestry of Indonesia map could be in the form agricultural land (food crops) and other productive land, so it is recommended to carry out screening first. Screening techniques can refer to the paper Jaung et al (2018) Sustainability 2018, 10(12), 4595 ( https://doi.org/10.3390/su10124595)

c. Multi-collinearity analysis should be carried out prior to Maxent analysis. So the variables included in the modeling are independent variables

Validity of the findings

The data (presence data) used is very small, so it is necessary to convey the limitations of the model. Discussions related to small samples in Maxent can refer to the paper Morales, Fernandes, Gonzales 2017 (https://doi.org/10.7717%2Fpeerj.3093)

Additional comments

First: I see that the critical land area is overestimated, because agricultural land, especially dry land/rain-fed agriculture, is included as degraded land (even though it is still productive as food production land). For example, in West Java, it is very visible that almost the entire province (except wetlands/rice fields) is categorized as critical land. So it is very important to carry out screening. For example, overlay with Land-use and Land cover Map.

Second: To implement research results, it seems necessary to differentiate between degraded land in Forest Areas (kawasan hutan) and outside forest areas (Other Use Areas/APL), so that recommendations will be more precise.

---

## Round 0.2 · accepted · Accept

As underlined by the reviewer, the Authors have significantly improved the manuscript, which is accepted in its present form.

Reviewer 2 ·

Basic reporting

no comment

Experimental design

no comment

Validity of the findings

no comment

Additional comments

The author has made significant improvements to the manuscript, especially on issues sensitive to the applicability of research results. The methodology has also been revised properly. As a reviewer, I appreciate the efforts made by the author.